# Extraction, Purification, and Structural Characterization of Polysaccharides from *Sanghuangporus vaninii* with Anti-Inflammatory Activity

**DOI:** 10.3390/molecules28166081

**Published:** 2023-08-16

**Authors:** Jinze Liu, Jinyue Song, Fusheng Gao, Weijia Chen, Ying Zong, Jianming Li, Zhongmei He, Rui Du

**Affiliations:** 1College of Traditional Chinese Medicine, Jilin Agricultural University, Changchun 130118, China; liujinze0602@126.com (J.L.); 17843031155@163.com (J.S.); m13180289293@126.com (F.G.); chenweijia_jlau@163.com (W.C.); zongying7699@126.com (Y.Z.); m15568781138@163.com (J.L.); 2Engineering Research Center for High Efficiency Breeding and Product Development Technology of Sika Deer, Changchun 130118, China

**Keywords:** *Sanghuangporus vaninii*, polysaccharide, ultrasonic extraction, optimization, structure analysis, antioxidant activity, anti-inflammatory activity

## Abstract

In order to obtain homogeneous *Sanghuangporus vaninii* polysaccharides with antioxidant and anti-inflammatory activities, a response surface method (RSM) was used to compare the polysaccharide extraction rate of hot water extraction and ultrasonic-assisted extraction from *Sanghuangporus vaninii*. The optimal conditions for ultrasonic-assisted extraction were determined as follows: an extraction temperature of 60 °C, an extraction time of 60 min, a solid–liquid ratio of 40 g/mL, and an ultrasonic power of 70 W. An SVP (*Sanghuangporus vaninii* polysaccharides) extraction rate of 1.41% was achieved. Five homogeneous monosaccharides were obtained by gradient ethanol precipitation with diethylaminoethyl–cellulose (DEAE) and SephadexG-100 separation and purification. The five polysaccharides were characterized by high performance liquid chromatography, the ultraviolet spectrum, the Fourier transform infrared spectrum, TG (thermogravimetric analysis), the Zeta potential, and SEM (scanning electron microscopy). The five polysaccharides had certain levels of antioxidant activity in vitro. In addition, we the investigated the anti-inflammatory effects of polysaccharides derived from *Sanghuangporus vaninii* on lipopolysaccharide (LPS)-induced RAW 264.7 cells and Kupffer cells. Further, we found that SVP-60 significantly inhibited the levels of pro-inflammatory cytokines, such as interleukin (IL)-1β, IL-6, and tumor necrosis factor (TNF)-α in lipopolysaccharide (LPS)-induced RAW 264.7 cells and promoted the level of the anti-inflammatory cytokine IL-10 in lipopolysaccharide (LPS)-induced RAW 264.7 cells. Our study provides theoretical support for the potential application of *Sanghuangporus vaninii* in the field of antioxidant and anti-inflammatory activities in vitro.

## 1. Introduction

Most diseases, including intestinal diseases, liver diseases, and so on, originate from inflammation [1,2]. Long-term local inflammation can cause systemic inflammation which, in turn, affects the local inflammation of previously unaffected organs [3]. In recent years, it has been found that the exacerbation of inflammation can be controlled by controlling the levels of pro-inflammatory molecules including interleukin-1β (IL-1β), interleukin-6 (IL-6), tumor necrosis factor α (TNF-α), and the anti-inflammatory factor interleukin-10 (IL-10) [4]. There are reports that medicinal fungal polysaccharides have certain levels of in vitro anti-inflammatory activity. For example, chaga extracts were shown to reduce NO production and release levels of TNF-α, IL-6, and IL-1β in lipopolysaccharide (LPS)-stimulated RAW 264.7 macrophages [5]. Lingzhi or Reishi medicinal mushrooms, *Ganoderma lucidum* (Agaricomycetes), restrained the production of NO, TNF-α, IL-1β, and IL-6 in lipopolysaccharide (LPS)-stimulated RAW 264.7 macrophages and was proven to be non-toxic in vitro [6].

*Sanghuangporus vaninii* (Ljub.) L.W.Zhou & Y.C.Dai (*S. vaninii*) is a large perennial medicinal fungus that is distributed in Japan, South Korea, and northeast China [7,8]. In China, it has been used as a traditional Chinese herbal medicine for more than 2000 years [9,10]. It is famous for rich concentrations of polysaccharides [11,12], triterpenoids [13], polyphenols [14], and other bioactive substances [15] and often used to treat a variety of female conditions, such as female bleeding and amenorrhea [9]. In addition, it was reported that Sanghuangporus sanghuang Mycelium has an inhibitory effect on excessive inflammation induced in mice [16]. *Sanghuangporus vaninii* has an inhibitory effect on breast cancer cells, and *Sanghuangporus vaninii* cultivated under forest conditions has a certain anti-cancer effect in human colon cancer cells [9,17].

In recent years, natural medicinal fungi have been used in the fields of drugs and nutraceuticals due to their biological activity [18]. Polysaccharides, as one of the most important active components in medicinal fungi, have received widespread attention. The extraction process plays a crucial role in the study of polysaccharides [19,20,21], because the extraction parameters of different extraction techniques have important effects on the yield, structure, and biological activity of polysaccharides [21,22,23]. Therefore, an effective extraction method is needed to obtain *S. vaninii* polysaccharides (SVPs).

In this study, five new polysaccharides, SVP-40, SVP-50, SVP-60, SVP-70, and SVP-80, were isolated from S. vaninii under gradient ethanol precipitation and optimized extraction conditions by the response surface method, and they were analyzed according to their molecular weights and monosaccharide compositions. UV, FT-IR spectroscopy and TG analysis were used to elucidate the structural characterization of polysaccharides. A morphological analysis was performed by SEM. In addition, the anti-inflammatory activity of SVP-60 was evaluated by measuring the degree of lipopolysaccharide (LPS)-induced inflammation in RAW 264.7 and Kupffer cells. This study can provide an experimental basis for the further study of SVP and certain guidance for the development of SVP foods or drugs.

## 2. Results

### 2.1. Single-Factor Experiment Analysis

#### 2.1.1. Influence of the Extraction Temperature on the SVP Extraction Yield

The extraction temperature is one of the factors affecting the extraction yield [24]. The effect of the extraction temperature on the SVP yield in the HWE process is studied. As shown in Figure 1A, the other extraction conditions are as follows: an extraction time of 3.0 h and a liquid–solid ratio of 30 mL/g. As shown in Figure 1A, when the extraction temperature was 80 °C, the extraction rate of SVP reached the highest level (0.60 ± 0.0351%). And, when the temperature exceeded 80 °C, the extraction rate of SVP decreased slightly and tended to be flat. The maximum yield was achieved at an extraction temperature of 80 °C. Therefore, 80 °C is considered to be optimal in this experiment. The effect of the ultrasonic temperature on the extraction yield of polysaccharides is shown in Figure 1a. The extraction process was performed using temperatures ranging from 40 °C to 80 °C. The other extraction conditions were as follows: an extraction time of 60 min, a liquid–solid ratio of 40 mL/g, and an ultrasonic power of 70 W. The temperature range was from 40 to 60 °C, and the yield was slightly decreased. The maximum extraction SVP yield (1.80 ± 0.1153%) from 40 to 80 °C occurred at a temperature of 60 °C. Therefore, 60 °C was selected as the optimal ultrasonic temperature for the UAE experiments.

#### 2.1.2. Effect of the Extraction Time on the Extraction Rate of SVP

The extraction time is another factor that affects the extraction efficiency and the selectivity of the extraction solution [25]. In the HWE process, the extraction time has a great influence on the yield of SVP. This is shown in Figure 1B. The extraction times are 1.0, 2.0, 3.0, 4.0, and 5.0 h. The other extraction parameters are as follows: a liquid–solid ratio of 30 mL/g, and an extraction temperature of 80 °C. The extraction rate increased from 1 h to 3 h with an increase in the extraction amount and reached the peak value at 3 h (0.57 ± 0.1114%). Considering the energy consumption and extraction cost, an optimal extraction time of 3 h is sufficient to obtain maximal polysaccharide extraction. During UAE, the extraction times were set to 30, 40, 50, 60, and 70 min. The other extraction parameters were as follows: a liquid–solid ratio of 40 mL/g, an extraction temperature of 60 °C, and an ultrasonic power of 70 W. As shown in Figure 1b, the extraction time increased from 30 to 60 min, and the extraction rate gradually decreased from 60 min. The optimal SVP yield was (1.54 ± 0.0153%). When the extraction time was 60 min, the polysaccharides in the broken cells were released during the initial extraction. And, the long extraction time induced polysaccharide degradation, leading to a decrease in the production of SVP. The optimal extraction time is 60 min.

#### 2.1.3. Effect of the Liquid–Solid Ratio on the Extraction Rate of SVP

In the HWE process, the liquid–solid ratio was set to 10, 20, 30, 40, and 50 mL/g, and the other influencing factors were as follows: an extraction temperature of 80 °C and an extraction time of 3 h. Figure 1C shows that the extraction efficiency was low at first, with a liquid–solid ratio of 10 mL/g. And, the extraction rate increased sharply at a liquid–solid ratio of 30 mL/g with a maximum value of 0.48 ± 0.0643%. Then, the extraction rate decreased as the liquid-to-solid ratio continued to rise, a phenomenon identical to that report by some [26]. Therefore, in the following experiments, a liquid–solid ratio of 30 mL/g was used to determine the effect of the liquid–solid ratio on the yield of SVP in HWE; the process is shown in Figure 1c. The extraction conditions were determined as follows: an extraction temperature of 60 °C, an extraction time of 60 min, and an ultrasonic power of 70 W. The SVP yield showed an increasing ratio of 20 to 40 mL/g, and the rising trend was similar to the HWE process. Based on the results shown in Figure 1c, 40 mL/g was used as the optimal liquid–solid ratio for subsequent experiments in the UAE process.

#### 2.1.4. Effect of the Ultrasound Power on the SVP Extraction Rate

Ultrasonic power is also an important parameter that influences the extraction of polysaccharides [27]. The effect of ultrasound on the extraction rate of SVP was studied under the conditions of an extraction temperature of 60 °C, an extraction time of 60 min, and a 40 mL/g liquid-to-solid ratio. As shown in Figure 1d, the extraction rate increased significantly with an increase in the ultrasonic dose and then decreased when the ultrasonic power exceeded 70 W. Therefore, the optimal extraction power is 70 W.

### 2.2. Optimization of SVP Extraction Parameters

#### 2.2.1. Statistical Analysis and Model Fitting

According to the results of the single-factor experiments and the Box–Behnken design experiment, the response surface method (RSM) was used to monitor the extraction characteristics of polysaccharide components in SVP in the HWE experiment. And, the optimal extraction conditions were determined. A total of 17 job designs were performed to optimize the three individual parameters in the current box [28]. The design can be applied to SVP hot water extraction. Data were analyzed using Design Expert 8.0.6.1, and the following polynomial equation was derived to express the yield as a function of the independent variable, denoting Y = +0.59 − 0.014 × A + 0.014 × B + 0.020 × C + 0.017 × A × B − 0.020 × A × C − 0.010 × B × C − 0.093 × A^2^ − 0.088 × B^2^ − 0.11 × C^2^, where Y is the predicted SVP production and A, B, and C are the liquid–solid ratio, extraction temperature, and extraction time. This was used to analyze the significance and applicability of the model and is summarized in the Table 1. As described, the model has an F-value of 61.69, and a low probability *p*-value (<0.0001), which indicates the high significance of the model. For the fit of the model, the coefficient of determination (R^2^) is 0.9875 and the adjusted coefficient is 0.9715, indicating that 97.15% of the model data could be explained by the model. The F-value with an insufficient fit is not significant (*p* > 0.05), which confirms the validity of the model. At the same time, the low coefficient of variation (CV) of 3.56 clearly indicates the reliability of the experimental value. Based on the data presented in Table 1, it is concluded that the order of the factors affecting the SVP extraction yield response value is as follows: extraction time > extraction temperature = liquid–solid ratio.

According to the results of the single-factor experiment and Box–Behnken design experiment, the response surface method (RSM) was used to monitor the extraction characteristics of polysaccharides from SVP in the UAE experiment, and the optimal extraction conditions were determined. A total of 29 Behnken designs were used to optimize the four individual parameters in the current box [29]. The design can be applied to the production of SVP ultrasonic-assisted extraction. Design Expert 8.0.6.1 software was used to analyze the data. The polynomial equation of the yield as a function of the independent variable is Y = +1.55 − 0.014 × A − 0.017 × B − 0.032 × C − 0.012 × D + 0.11× A × B + 0.000 × A × C + 0.054 × A × D + 0.055 × B × C + 0.038 × B × D − 0.060 × C × D − 0.10 A^2^ − 0.098 × B^2^ − 0.21 × C^2^ − 0.22 × D^2^, where Y is the predicted SVP yield and A, B, C, and D are the extraction time, extraction temperature, extraction power, and liquid–solid ratio, respectively. This was used to analyze the significance and applicability of the model. The summary is shown in Table 2. As mentioned above, the F-value of the model is 125.97, and the low probability *p*-value (<0.0001) indicates the high significance of the model. The coefficient of determination (R^2^) of model fit is 0.9921, and the coefficient of adjustment is 0.9842, indicating that 98.42% of the model data could be interpreted by the model. The F-value of inadequate fitting is not significant (*p* > 0.05), which confirms the validity of the model. Meanwhile, the low coefficient of variation (CV) of 1.48 clearly indicates the reliability of the experimental values. According to the data presented in Table 2, the sequence of factors affecting the response value of the SVP extraction rate is as follows: extraction power > extraction temperature > extraction time > liquid–solid ratio.

#### 2.2.2. Response Surface Diagram

The 2D and 3D response surface curves are used to determine the relationship for each dependent variable [30]. The two-dimensional and three-dimensional response surface curves for hot water extraction are shown in Figure 2. Figure 2A,a shows the interaction of the solid–liquid ratio and the extraction temperature on the SVP yield when the extraction time is 3 h. In the HWE process, it was found that the extraction rate of SVP increased and then decreased with the increase in the solid–liquid ratio. Figure 2B,b shows the effects of the extraction time and liquid–solid ratio on the SVP yield. When the extraction temperature was 80 °C, the yield of SVP first increased and then decreased with the increase of extraction time and liquid–solid ratio. The interaction between the extraction time and extraction temperature was studied as shown in Figure 2C,c. When the liquid–solid ratio was 30 mg/mL, the yield of SVP first increased and then decreased with increases in the extraction time and extraction temperature. Figure 3A–F and Figure 4A–F show the interactions between the extraction variables (extraction temperature, extraction time, liquid–solid ratio, and effect of ultrasonic power) and the extraction rate of SVP in the UAE process. The extraction temperature, extraction time, liquid–solid ratio, and ultrasonic power can significantly affect the extraction rate of SVP. With increases in these, the extraction rate increases significantly. When one of the variables is fixed, three variables in a certain range usually increase slowly. When these variables exceed a certain value, a steady trend or gradual decline is shown. In addition, it can be found that, in the UAE process, the solid–liquid ratio has a weak influence on the yield of polysaccharides, and the conclusion is shown in Table 2. 

### 2.3. Optimization and Validation

We used Design Expert 8.0.6 software, which is mentioned in various reports and is based on the response surface and variance analysis. The optimal extraction conditions of SVP in the HWE process could be predicted as an extraction temperature of 80.65 °C, an extraction time of 3.09 h, a solid–liquid ratio of 29.22 g/mL, and a maximum yield of 0.961. The optimal UAE conditions are an extraction temperature of 57.34 °C, an extraction time of 57.70 min, a solid–liquid ratio of 1:39.36 g/mL, and an ultrasonic power of 68.9 W. Under these conditions, the SVP yield can reach the theoretical maximum of 1.55592%. In order to verify the adequacy of the model equation, three experiments were conducted, and their average values were calculated. In the production practice, the optimum parameters of the HWE process were adjusted as follows for ease of operation: an extraction temperature of 80 °C, an extraction time of 3.00 h, and a solid–liquid ratio of 1:30 g/m L. The optimal parameters of UAE were adjusted as follows: an extraction temperature of 60 ◦C, an extraction time of 60 h, a solid–liquid ratio of 40 g/mL, and an ultrasonic power of 70 W. Under these conditions, the experimental extraction yield of HWE is 0.83 ± 0.12% (*n* = 3), and that of UAE is 1.41 ± 0.11% (*n* = 3), which is in good agreement with the predicted results. The experiment fully verifies the correctness and reliability of the extraction model.

### 2.4. Isolation and Purification of SVP

The crude polysaccharides SVP-40, SVP-50, SVP-60, SVP-70, and SVP-80 extracted from the dried powder of *S. vaninii* accounted for about 0.35%, 0.39%, 0.77%, 0.26%, and 1.08% of the dried plant mass, respectively (Table 3). As shown in Figure 5A, the elution of the SVP with 40% ethanol precipitation was carried out by DEAE-52 cellulose column chromatography with NaCl solution. And, an acidic fraction SVP-40 was obtained after anion exchange chromatography. Figure 5B shows the polysaccharide of SVP-40 obtained by glucan gel G-100, demonstrating the presence of a homogeneous polysaccharide. Purified polysaccharide precipitated with 50% ethanol in SVP (Figure 6A,B), purified polysaccharide precipitated with 60% ethanol in SVP (Figure 7A,B). However, as shown in Figure 8A, the eluted polysaccharide SVP-70 precipitated with 70% ethanol was used for SVP to obtain a neutral polysaccharide. In Figure 8B, a homogeneous polysaccharide is obtained with G-100 glucan. And purified polysaccharide precipitated with 80% ethanol in SVP (Figure 9A,B) showed the same results as those of SVP-40. 

As shown in Table 3, the purified SVP-40, SVP-50, SVP-60, SVP-70, and SVP-80 were determined by the phenol–sulfuric acid method to have total sugar contents of 85.53%, 86.90%, 92.40%, 90.12%, 85.65%, and 0.29%, respectively. The contents of protein and uronic acid were 0.29%, 0.19%, 0.91%, 2.43%, and 0.32% and 6.03%, 7.87%, 4.39%, 0.95%, and 0.47%, respectively.

### 2.5. Monosaccharide Composition, Molar Ratio, and Molecular Weight of SVP

As shown in Table 3, after monosaccharide derivation by the PMP method, the monosaccharide compositions of SVP-40, SVP-50, SVP-60, SVP-70 and SVP-80 were analyzed as derivatives using HPLC (high performance liquid chromatography). The HPLC figure is shown in Figure 10. In SVP-40, Glu (13.7 mol%) is the main monosaccharide, followed by Man (2.2 mol%) and GalA (8.5 mol%). Small amounts of Rham (1.2 mol%) and GalA (1.8 mol%) were detected in SVP-50, and the main monosaccharide was Man (5.5 mol%). SVP-60, which is composed of Man (1.4 mol%), Rham (1.1 mol%), GluA (0.5 mol%), GalA (0.6 mol%), Glu (2.0 mol%), and Ara (1.8 mol%) contains the most monosaccharides among all polysaccharide grades. The neutral polysaccharide SVP-70 is mainly composed of Glu (2.7 mol%). The acidic heteropolysaccharide SVP-80 is mainly composed of Rham. The standard curve was formed by taking glucan with different relative molecular weights as the standard. Y = −0.814x + 9.8883 (R^2^ = 0.9957). The molecular weights of SVP-40, SVP-50, SVP-60, SVP-70, and SVP-80 were determined to be 295.06 k, 215.36, 245.09, 273.75, and 274.77 k Da, respectively.

### 2.6. Uv-vis Spectral Characteristics

As shown in Figure 5C, Figure 6C, Figure 7C, Figure 8C and Figure 9C, the five SVP polysaccharides showed their maximum absorption peaks at about 200 nm, which is related to the fact that they are polysaccharides. No obvious absorption peaks were found at 260 nm and 280 nm, indicating that the protein contains no nucleic acid or protein [31].

### 2.7. Fourier Transform Infrared Spectral Characteristics

The FT-IR spectra of the five SVP polysaccharides (Figure 5D, Figure 6D, Figure 7D, Figure 8D and Figure 9D) are similar, showing characteristic absorption of typical polysaccharides. In the case of SVP-60, functional groups were monitored in the range of 500–4500 cm^−1^ by FT-IR studies (Figure 7D). Therefore, the wide absorption peak at 3313 cm^−1^ is related to the O-H hydroxyl group [32], and there is stretching vibration in the sugar ring. The peak of about 2930 cm^−1^ indicates the presence of tensile vibration and bending vibration in the C-H bond [33]. Furthermore, a peak occurs at around 1637 cm^−1^ and is assigned to C=O tensile vibration [34]. A peak near to 1090 cm^−1^ indicates a possible pyran ring in the composition. The absorption peak at around 1036 cm^−1^ is caused by the C-O-C tensile vibration [35].

### 2.8. Results of the TG Analysis

TG curves usually show that the thermal stability of materials is different due to their closely related structural and morphological characteristics. To some extent, the suitability of polysaccharides depends on their thermal properties. Due to the water evaporation from the five purified polysaccharides of SVP, thermal decomposition occurred in the molecular interior. The thermal characteristics of five SVP polysaccharides at 30–800 °C are shown in Figure 11A. TG curves show that SVP-40 is more stable than other components below 200 °C. SVP-60 is the second most stable, and SVP-80 has the strongest thermal degradability. These differences indicate that the monosaccharide composition, water content, molecular aggregation, and structural morphology of the five components are different. At 800 °C, when the polysaccharide decomposition process is complete, the residual weights of SVP-40, SVP-50, SVP-60, SVP-70, and SVP-80 are 23.84%, 17.17%, 9.67%, 25.88%, and 14.68%, respectively. Obviously, the order of thermal stability is SVP-70 > SVP-40 > SVP-50 > SVP-80 > SVP-60, which may be related to the types and quantities of monosaccharides in the five polysaccharides, and the specific reasons for these results need further study. In conclusion, the five polysaccharides of SVP have good thermal stability. SVP-70, in particular, is expected to be used as an additive in the food industry.

### 2.9. Zeta Potential

As shown in Figure 11B, the five components are anionic polysaccharides. The absolute potential values of SVP-80 and SVP-60 are larger than those of the other three components. We think that this may be because they contain a certain amount of uronic acid and produce more anions. Overall, the five polysaccharides of SVP are considered to have more stability.

### 2.10. Antioxidant Activities In Vitro

#### 2.10.1. DPPH Radical Scavenging Capacity

The DPPH method is widely used to evaluate the antioxidant capacity of natural products in vitro due to its simple operation and good reproducibility. The DPPH radical alcohol solution is dark purple in color at 517 nm and changes color once it interacts with the polysaccharide. The greater the antioxidant concentration in the polysaccharide, the lighter its color. The scavenging results of the five SVP polysaccharides with the DPPH free radical are shown in Figure 12A. The five kinds of SVP polysaccharides have certain scavenging effects on the DPPH free radical. Within the experimental concentration range, the scavenging ability of SVP on DPPH free radical increased gradually with the continuous increase in the polysaccharide concentration, showing a certain dose effect dependence. The free radical scavenging ability of SVP-80 for DPPH increased first, and then decreased and gradually increased. As the polysaccharide concentration climbed to 1 mg/mL, the scavenging rates of the DPPH free radical were 70.58 ± 0.72%, 56.73 ± 3.83%, 65.63 ± 2.17%, 61.84 ± 0.64, and 63.94 ± 0.98% for SVP-40, SVP-50, SVP-60, SVP-70, and SVP-80. Obviously, the DPPH radical scavenging activity of SVP-40 was the highest in the range of concentrations tested, and there were no significant differences in the clearance abilities of the five polysaccharides.

#### 2.10.2. Hydroxyl Radical Scavenging Ability

The hydroxyl radical is a kind of free radical with a strong oxidizing ability, and it has been found in related literature to cause some harm to the body, such as aging effects and cancer. As shown in Figure 12B, when the polysaccharide concentration was increased from 0 mg/mL to 1.0 mg/mL, the scavenging ability of the five SVP polysaccharides on hydroxyl radicals showed a significant upward trend in a dose-dependent manner, as described above. At 1.0 mg/mL, the hydroxyl radical scavenging capacities of SVP-80, SVP-70, SVP-60, SVP-50, and SVP-40 were 56.47 ± 10.55%, 39.58 ± 0.72%, 59.76 ± 3.71%, 60.93 ± 1.23%, and 66.72 ± 3.12%. Obviously, SVP-40 had the strongest hydroxyl radical scavenging ability. Surprisingly, its scavenging ability increased first and then decreased, and it was followed by SVP-50.

#### 2.10.3. Superoxide Anion Radical Scavenging Ability

The scavenging ability of the superoxide anion radical is one of the key indicators of antioxidant activity. This is not only because the superoxide anion itself causes cell and organ damage, but also because the superoxide anion, like hydroxyl radicals, is associated with some major diseases, such as aging effects and cancer. Figure 12C shows that the scavenging abilities of the five SVP polysaccharides and VC are closely related to their concentrations. At a concentration of 1 mg/mL, the superoxide free radical scavenging rate of SVP-80 was 70.49 ± 0.14%, that of SVP-70 was 68.33 ± 5.49%, and that of SVP-60 was 71.24 ± 1.88%. That of SVP-50 was 71.18.6 ± 1.13%, that of SVP-40 was 74.54 ± 2.01%, and that of VC was 97.79 ± 0.51%. Compared with other radicals, SVP-40 had the highest scavenging ability against superoxide free radicals, followed by SVP-60. And, there was no significant difference in the clearance abilities of the five polysaccharides.

#### 2.10.4. Determination of the Reducing Power

The antioxidant capacity of polysaccharides refers to the reducing ability of polysaccharides to remove free radicals: the greater the reducing power, the greater the oxidation resistance. The reducing power is also often a measure of a compound’s antioxidant capacity. Figure 12D shows that the scavenging abilities of the five SVP polysaccharides and VC were closely related to the concentration in a concentration-dependent manner. Five kinds of SVP polysaccharides had certain reducing abilities. At a concentration of 1 mg/mL, the scavenging rate of SVP-40 was 62.31 ± 2.11%, that of SVP-50 was 69.35 ± 1.00%, that of SVP-60 was 68.73 ± 2.8%, and that of SVP-70 was 59.14 ± 0.63%. That of SVP-80 was 76.99 ± 1.45% and that of VC was 96.84 ± 0.64%. Compared with other polysaccharides, SVP-80 had the highest reducing power, followed by SVP-50 and SVP-60, while SVP-70 had the lowest reducing power.

### 2.11. Congo Red Experimental Analysis

Congo red is an acidic dye that binds to triple helical chain structures to form complexes. As shown in Figure 13A, the UV absorption levels of the four polysaccharides, SVP-40, SVP-50, SVP-70, and SVP-80, were shifted to the long-wave direction with an increasing NaOH concentration, indicating that the four samples formed complexes with the triple helix structure. With an increase in the NaOH concentration, the maximum absorption wavelength decreased, proving the disintegration of the triple helix structure. Therefore, it is believed that these four polysaccharides have triple helix structures. However, the wavelength change of SVP-60 was not obvious enough, so it is considered that there was no reaction between the two. Therefore, whether SVP-60 has a triple helix structure needs to be further explored.

### 2.12. SEM Electron Microscopy Analysis

As can be seen from the appearance morphologies scanned by B, C D, E, and F in Figure 12, the five polysaccharides are mostly clustered in a sheet structure and are clustered together with small fragment structures and more gaps between dispersed fragments. Moreover, it is believed that the five polysaccharides have rod-like structures.

### 2.13. Anti-Inflammatory Activity Analysis

#### 2.13.1. Effects of Five Purified Polysaccharides on the Viability of RAW 264.7 Macrophages and Kupffer Macrophages

To illustrate the potential cytotoxicity of polysaccharides, we evaluated the cytotoxicity of five SVP polysaccharides on two different cell lines: RAW 264.7 macrophages and Kupffer macrophages. As shown in Figure 14B–F and Figure 15B–F, we found that five polysaccharides inhibited the activity of RAW 264.7 and Kupffer cells at large concentrations. Surprisingly, SVP-50 had a proliferative effect on RAW 264.7 cells at concentrations of 10–20 μg/mL, and the cell survival rate reached 116.25 ± 3.42% at 20 μg/m L. SVP-70 also had a proliferative effect on Kupffer cells at concentrations of 5–20 μg/mL, and the cell survival rate reached 131.19 ± 3.42% at 5 μg/m L. At a concentration of 5 μg/m L, the cell survival rate reached 131.19 ± 6.11%. Based on the above results, SVP-60 was found to be of the highest purity with a weak cell proliferation ability and little effect on the cell viability. Therefore, SVP-60 was selected as the target for the evaluation of the anti-inflammatory ability of SVP polysaccharides.

#### 2.13.2. Effects of the Purified Polysaccharide SVP-60 on the Viability of LPS-Induced RAW 264.7 Macrophages

Most of the purified polysaccharides have significant anti-inflammatory activity. LPS has been found to be a good drug to establish in vitro anti-inflammatory activity in most literature reports. However, it has been also found that LPS has a bidirectional effect and does not inhibit the growth of macrophages when the LPS concentration reaches a certain level. On the contrary, it may trigger an immune cell response and increase the proliferation of cells. Twenty-four hours after modeling, LPS had no inhibitory effect on RAW 264.7 cells at concentrations of 0.05–0.5 μg/mL, but it had a proliferation effect without significant differences, and the cell activity was significantly decreased at concentrations of 1–1.5 μg/mL (*p* < 0.001). The cell viability levels were 64.05 ± 4.22% and 65.17 ± 3.83%, respectively, as shown in Figure 14A. LPS (1 μg/mL) was selected as the model. Compared with the control group, the cell viability of the model group was significantly decreased (*p* < 0.01). Different SVP-60 concentrations (50–70 μg/mL) were used to treat the RAW 264.7 cells. Compared with the control group, the cell viability of the model group decreased significantly, and the cell viability decreased to 59.58 ± 13.03%. The RAW 264.7 cells were treated with different concentrations of SVP-60 (50–70 μg/mL), which were 81.12 ± 6.35%, 78.60 ± 12.43%, and 86.00 ± 2.69%, respectively. As shown in Figure 14G, the cell proliferation capacity was increased after SVP-60 treatment compared to the model group. However, there was a difference only in RAW 264.7 cells treated with a high dose of 70 μg/mL (*p* < 0.05).

#### 2.13.3. Effects of the Purified Polysaccharide SVP-60 on the Viability of LPS-Induced Kupffer Macrophages

An LPS concentration of 1 μg/mL was selected as a model, as shown in Figure 15A. Figure 15G shows that, compared with the model group (67.19 ± 8.96%), the activity of Kupffer cells in the administration group was significantly increased by 82.34 ± 3.54%, 79.76 ± 4.01%, and 87.16 ± 2.90%, respectively (*p* < 0.01). Surprisingly, there was no dose-dependent increase. The effects of SVP-60 on the LPS-induced RAW.264.7 and Kupffer cell viability suggest that the SVP polysaccharide has a certain level of anti-inflammatory activity in vitro.

#### 2.13.4. Effect of SVP-60 on the Production of LPS-Induced Pro-Inflammatory and Anti-Inflammatory Factor in RAW 264.7 Cells

To further determine the anti-inflammatory activity of SVP-60 in vitro, we measured the levels of proinflammatory cytokines, such as TNF-α, IL-1β, and IL-6, and the anti-inflammatory factor IL-10, released by macrophage RAW 264.7, following stimulation by LPS. In response to LPS stimulation, macrophages released pro-inflammatory cytokines, such as TNF-α, IL-1β, and IL-6. RAW 264.7 cells were treated with LPS in the presence or absence of SVP-60, and the levels of TNF-α, IL-1β, and IL-6 were measured by ELISA. As shown in Figure 16A–C, SVP-60 suppressed the production of TNF-α, IL-1β, and IL-6 in LPS-induced RAW 264.7 cells in a concentration-dependent manner. Compared with the LPS group, TNF-α levels decreased by 12.25%, 21.42%, and 36.38% at different concentrations of SVP-60 (Figure 16A). The expression of IL-6 in macrophages increased to 223.08 ± 8.902 pg/mL after stimulation with LPS. However, these increases were significantly reduced when SVP-60 was added at concentrations of 50, 60, and 70 μg/mL (*p* < 0.0001). Compared with the LPS group, different concentrations of SVP-60 reduced the IL-6 level released by RAW 264.7 cells by 60.43%, 69.59%, and 75.98%, respectively (Figure 16B). Our data show that the expression of IL-1β was significantly increased in the LPS-stimulated group compared with the control group (Figure 16C). At different concentrations of SVP-60, there was a significant inhibitory effect on the IL-1β level compared to the LPS group (*p* < 0.001). For example, 70 μg/mL reduced the IL-1β level by 48.35% compared to the LPS group.

It is well-known that macrophages also release the anti-inflammatory factor IL-10 when stimulated by LPS. RAW 264.7 cells were treated with LPS in the presence or absence of SVP-60, and the level of IL-10 was detected by ELISA. As shown in Figure 16D, the ELISA level of IL-10 was significantly higher in the LPS group than in the control group. Compared with the LPS group, 50 μg/mL and 60 μg/mL of SVP-60 reduced the IL-10 level by 9.56% and 4.77%, respectively. And, the IL-10 level of RAW 264.7 was significantly increased by 16.04% after treatment with 70 μg/mL of SVP-60.

## 3. Discussion

Many species of medicinal fungi have been used as foods, medicines, and nutraceuticals for more than 2000 years [18]. *S. vaninii* is recognized for its medicinal properties and is used to treat various diseases in Asia. Early scientific reports investigated the biological activity and toxic effects of *S. vaninii* from different regions [10,36]. However, previous studies have never used *S. vaninii* from Jilin Province, China. Polysaccharides are believed to have many biological activities, including antioxidant and immune inflammatory activities. *S. vaninii* has a high content of polysaccharides, but the extraction rate has not been able to increase. Our study was the first to determine the SVP with the highest extraction rate under optimal conditions through a single-factor combined response surface methodology. Five polysaccharides were isolated and purified using ethanol gradient precipitation, and it was further confirmed that SVP-60 is an acidic polysaccharide containing Man, Rham, GluA, GalA, Glu, and Ara. Through a relevant analysis of structural characterization, it was determined that all five polysaccharides have a certain level of stability. The aim of the present study was to obtain a more efficient extraction method for SVP and to determine its biological activities, including its antioxidant and immune anti-inflammatory activities. 

At present, medical science believes that the pathological processes of various diseases are due to the excessive oxidation of oxygen free radicals, which damages cells [37]. In recent years, the antioxidant effects of polysaccharides have received extensive attention worldwide, and there is a growing desire to find natural and effective antioxidants. Polysaccharides isolated from ginseng have been reported to have scavenging activity against superoxide radicals, hydroxyl radicals, and ABTS radicals [38]. In this study, the antioxidant activity of five SVP polysaccharides was determined in vitro by antioxidant experiments, including the DPPH radial scavenging ability, hydroxyl radial scavenging ability, superoxide anion radical scavenging ability, and reducing ability. These findings further demonstrate that the in vivo biological activity of *S. vaninii* needs to be further explored in terms of the specific antioxidant indicators SOD and CAT [39] and the related protein pathway Nrf-2/Keap-1 [40]. 

Inflammation remains one of the major health concerns globally, and the etiologies of various diseases, such as Escherichia coli disease [41], hepatitis [42], and cancer [43], have been found to be closely related to inflammation. Inflammation is a complex process that is triggered by a variety of factors and is a natural biological response to tissue damage. Under normal circumstances, the occurrence of moderate inflammation is conducive to the healing of human tissues. However, it can also be harmful. Regulating the production of inflammatory cytokines, such as tumor necrosis factor α (TNF-α), interleukin-6 (IL-6), and interleukin-1β (IL-1β), as well as the anti-inflammatory factor interleukin-10 (IL-10), has been reported to be a key mechanism for controlling inflammation [5,44,45]. Therefore, monitoring the expression levels of these anti-inflammatory and pro-inflammatory factors is crucial for us to further understand inflammation. RAW 264.7 and Kupffer macrophages are excellent models for anti-inflammatory drug screening and the subsequent evaluation of pathway inhibitors leading to the induction of proinflammatory cytokines. In addition, we elucidated the regulatory role of SVP-60 in the production of inflammatory cytokines in LPS-stimulated RAW 264.7 cells in vitro.

Firstly, the effects of five polysaccharides, SVP-40, SVP-50, SVP-60, SVP-70, and SVP-80, on the viability of RAW 264.7 and Kupffer macrophages were studied, and the results showed that they were basically non-toxic. Previous articles reported that, after LPS treatment, the cell viability of RAW 264.7 may change due to the release of corresponding inflammatory substances, which may decrease, increase, or remain unchanged. However, in our study, there was no significant change after LPS (3 μg/mL) treatment compared to untreated cells. Therefore, 1 μg/mL LPS was selected as the model group in this experiment. We evaluated the production of tumor necrosis factor α (TNF-α), interleukin-6 (IL-6), and interleukin-1β (IL-1β), as well as levels of the anti-inflammatory factor interleukin-10 (IL-10), and found that SVP-60 inhibited the production of IL-1β, IL-6, and TNF-α in LPS-induced RAW 264.7 and promoted the secretion of the anti-inflammatory factor IL-10. These findings further demonstrate the anti-inflammatory activity of SVP in vitro, but do not elucidate the underlying mechanism of several intracellular signaling pathway proteins (MAPKs, NF-kB, Akt) [46,47]. Therefore, further investigation needs to be conducted to determine whether SVP-60 can regulate the anti-inflammatory protein signaling pathway related proteins to reduce inflammation in the future.

## 4. Materials and Methods

### 4.1. Materials

The medicinal fungus (*S. vaninii*) was harvested in June 2022 from Hongshi Forestry Bureau, Huadian City, Jilin Province (42°97′ N/126°74′ E) and was identified by the director of the Chinese Society of Fungi Haiying Bao, School of Traditional Chinese Medicine, Jilin Agricultural University (Changchun, China). It is identified as Jilin Agricultural University herbal medicine preservation sample number 2022-06-198. The selected *S. vaninii* was dried at 40 °C and later stored at −20 °C after being crushed and passed through an 80 mesh sieve.

### 4.2. Extraction Method

A total of 200 g SVP is taken. SVP was degreased after immersion with sufficient petroleum ether and later re-fluxed three times by adding 80% ethanol (W/V = 1:5) to remove oligosaccharides, small molecules, and some colored substances, and the remaining SVP was dried naturally and set aside [48].

#### 4.2.1. HWE Procedure

A total of 2.0 g of the pretreatment sample was extracted with distilled water (liquid–solid ratio range 1:10 to 1:50 g/mL). The temperature of the water bath was at the given temperature (within ±1.0 °C, extraction temperature range of 60 to 100 °C). The extraction time range was 1 to 5 h. The extracted material was precipitated over eight layers of medical gauze, and the supernatant was taken. After extraction, the extract was blended and concentrated under vacuum using a rotary evaporator at 65 °C and concentrated to one-fifth of the initial volume. The concentrate was added to anhydrous ethanol to obtain a final concentration of 80% (*v*/*v*) and incubated in a refrigerator at 4 °C for 12 h. The precipitate was collected and dried to obtain crude polysaccharides. The percentage of SVP extraction (%) is given by the formula Y (%) = W_1_/W_2_ × 100%, where W_1_ is the polysaccharide weight after drying, and W_2_ is the initial dry weight of the sample [21].

#### 4.2.2. UAE Procedure

Ultrasound-assisted extraction was performed in an ultrasonic cleaning tank (Model KH-300DB, Jiangsu Kunshan Ultrasonic Instruments Co., Ltd., Kunshan, China). A total of 2.0 g of each pretreatment sample (liquid to solid ratio range from 20 to 60 g/mL) was extracted with distilled water. The extraction temperature range was 40 to 80 °C, the ultrasonic power range was 40 to 80 W, and the extraction time range was 30 to 70 min. The other steps were followed as described above.

### 4.3. Box–Behnken Design (BBD)

The HWE conditions of the SVP were further optimized using RSM based on single-factor experiments [49,50]. Three independent variables (A, liquid-to-solid ratio; B, extraction temperature; and C, extraction time) were used at three levels. Table 4 gives the ranges of the independent variables, the levels of the independent variables, and the results of the entire design consisting of 17 experimental sites performed in random order; these conditions are based on the results of preliminary experiments. For the statistical calculations, each variable was coded at three levels: −1, 0, and 1. RSM was also used to optimize the SVP UAE conditions. Four independent variables (A, extraction time; B, extraction temperature; C, liquid-to-solid ratio; and D, ultrasonic power) were used at three levels. The complete design consisted of 29 experiments, including 24 factorial experiments and five replicates at the center point. All experiments were performed randomly, as shown in Table 5.

### 4.4. Separation and Purification of S. vaninii Polysaccharide with Different Ethanol Gradients

A total of 200 g of defatted SVP was extracted according to the optimal method presented in Section 2.3 and then filtered through multiple layers of gauze. The filtrate was collected and centrifuged (3000 rpm, 5 min). The supernatant was concentrated under reduced pressure to one-tenth of the original volume and then precipitated with ethanol to obtain a final concentration of 40% at 4 °C for 24 h. After centrifugation, the precipitate was collected and freeze-dried in a freeze-dryer to obtain the herbal polysaccharide SVP-40. Ethanol was then added to the supernatant to obtain a final concentration of 50%, placed at 4 °C overnight, centrifuged, and then the precipitate was collected and freeze-dried to obtain the polysaccharide SVP-50. According to this method, the concentration of ethanol was increased sequentially to obtain the polysaccharides SVP-60 and SVP-70, and SVP-80 [51]. Each of the five polysaccharides was dissolved in distilled water, decolorized with macroporous resin, deproteinated by the Sevag method, and dialyzed with deionized water using a dialysis bag (MW3500) for 72 h. The dialysate was collected and precipitated with 40%, 50%, 60%, 70%, and 80% ethanol sequentially and subjected to overnight centrifugation at 4 °C. Finally, the initially purified polysaccharides, SVP-40, SVP-50, SVP-60, SVP-70, and SVP-80, were obtained and stored in a desiccator for later use.

SVP-40, SVP-50, SVP-60, SVP-70, and SVP-80 were used to purify different concentrations of polysaccharides by gradient elution with NaCl solution (0–1 M) on a DEAE-52 cellulose column (2.4 cm × 40 cm) at a rate of 1 mL/min [52]. The main obtained fractions were loaded onto a Sephacryl™ S-100 column (2.4 cm × 40 cm) and further purified with water at a flow rate of 0.5 mL/min. The main polysaccharide components were prepared by dialysis and freeze-drying.

### 4.5. Determination of the Chemical Composition and Monosaccharide Compositiont

The total sugar content [53] was determined by the phenol sulfate method using glucose as a standard (0.01–0.06 mg/mL). The protein content was determined according to the method of Yang et al. (0.025–0.25 mg/mL) [54]. The content of uronic acid was estimated by the m-hydroxybiphenyl method using galacturonic acid as a standard (0.01–0.06 mg/mL).

The monosaccharide compositions of five polysaccharides were determined according to Tang’s method with slight modifications [55]. In brief, samples were hydrolyzed at 110 °C using 2 M sulfuric acid in 1 mL distilled water for 6 h and then adjusted to pH = 7 using 4 mol/L NaOH, followed by Agilent tc-c18 (4.6 × 250 mm, 5 μm) for HPLC analysis. Calibration curves were prepared from PMP-derived standards including arabinose (Ara), rhamnose (Rha), fucose (Fuc), xylose (Xyl), mannose (Man), galactose (Gal), and glucose (Glc).

### 4.6. Determination of the Molecular Weight

The homogeneity and average molecular weight (Mw) (1 mg/mL, 10 μL) of the samples were determined by HPGPC passing through a column (Tskgel-g3000pwxl) on a Waters evaporative light scattering detector ELSD [56]. Glucan polymers with different Mw (180, 270, 4300, 5250, 9750, 13,050, 36,800, 64,650, 135,350, 300,600, 2,000,000 Da) were used to prepare the calibration curves. The retained volume was converted to the molecular weight according to the calibration curve provided by the above standard. The sample was the same as the standard.

### 4.7. Uv-vis Spectral Analysis

SVP-40, SVP-50, SVP-60, SVP-70, and SVP-80 were dissolved in distilled water to prepare 1 mg/mL of polysaccharide solution and scanned by a UV spectrophotometer (UV-6100, Mapada) in the range of 190–500 nm. The polysaccharide was tested for protein (280 nm) and nucleic acid (260 nm) conjugates [57].

### 4.8. FT-IR Spectral Analysis

A total of 2 mg of each dried sample (SVP-40, SVP-50, SVP-60, SVP-70, SVP-80) was placed on a Perkinelmer instrument to record the FT-IR spectra. FT-IR spectra were recorded from 4000 to 500 cm^−1^ with a Spectrum Two infrared spectrometer [58,59].

### 4.9. TG Analysis

The TG analysis was performed in the temperature range of 30 to 800 °C, with heating rates ranging from 0.1 °C/min to 150 °C/min. Using the Mettler Tolidor TGDSC^3+^ instrument, the scanning speed was 1 °C/min in an air environment [60].

### 4.10. Zeta Potential

The higher the absolute value of the Zeta potential (positive or negative), the more stable the material system is, and the lower the Zeta potential is, the less stable the material is. In this experiment, the Brookhaven Instrument Company Omni machine was used to measure the Zeta potential [61,62].

### 4.11. Determination of the Antioxidant Activity In Vitro

The activity was determined by using previously described methods [63,64]. The antioxidant activity of purified SVPs was evaluated by measuring the reducing power and the DPPH radical [65], hydroxyl radical, and superoxide anion radical scavenging capacities [66]. The determination of the reducing power of SVP-40, SVP-50, SVP-60, SVP-70, and SVP-80 was conducted using previously described methods. Vitamin C was used as a positive control.

### 4.12. Congo Red Experiment on Purified Polysaccharides

According to the method used in the experiment with slight modifications, we took 1 mL of Congo red solution (80 μM) and mixed it with 1 mL of SVP-40, SVP-50, SVP-60, SVP-70, and SVP-80 (1 mg/mL). Then, the UV wavelength was 400–600 nm. The absorption wavelength of the mixed solution was determined in different concentrations of NaOH (0, 0.1, 0.2, 0.3, 0.4, 0.5 mol/L) [67,68].

### 4.13. Detection of Purified Polysaccharide by Scanning Electron Microscopy (SEM)

An appropriate amount of the SVP sample was spread on conductive silica gel, sprayed with gold, placed on a SU8020 electric scanner, and recorded. The working voltage was set to 5.0 kV, and the solid shape of the sample was observed at different magnification appearances [69].

### 4.14. Cell Culture

#### 4.14.1. Cell Viability Test

RAW 264.7 macrophages and Kupffer macrophages were cultured in DMEM containing fetal bovine serum (10%), penicillin (100 IU/mL), and streptomycin (100 mg/L) at 37 °C and 5%CO_2_. A total of 100 μL of the cell supernatant (1 × 10^4^ cells per well) was inoculated on 96-well plates and incubated for 24 h. Different 100 μL samples (SVP-40, SVP-50, SVP-60, SVP-70, SVP-80) and different concentrations (1, 5, 10, 20, 40, 50, 60, 70, 80, 100, 500, 1000 μg/mL) were cultured for 24 h. We added 100 μL CCK-8 per well and incubated for 1 h. The cytotoxic effects of all samples were evaluated. Each dose had three repetitions.

#### 4.14.2. Anti-Inflammatory Activity of Macrophages

To evaluate the anti-inflammatory activity of SVP polysaccharides against RAW 264.7 and Kupffer cells, CCK-8 was used to quantify the cell viability. The cell activity of LPS at different concentrations (0.05, 0.1, 0.5, 1, 1.5, 2, 2.5, 3 μg/mL) was detected by CCK-8, and 1 μg/mL LPS was selected as the modeling concentration.

#### 4.14.3. Measurement of IL-6, IL-10, TNF-α, and IL-1β

RAW 264.7 cells (1 × 10^6^ cells/well) were pretreated with different concentrations of SVP-60 for 1 h and then stimulated with LPS (1 μg/mL) for 24  h. The concentrations of IL-6, IL-10, TNF-α and IL-1β were assayed using ELISA kits according to the manufacturer’s instructions. 

### 4.15. Statistical Analysis

All figures were formed using Graphpad Prism 8.0.2. Data are expressed as the mean ± standard deviation (SD). One-way analysis of variance (ANOVA) was used for the statistical analysis of data, followed by Tukey’s post-hoc multiple comparison test. Statistical significance is defined as (*) *p* < 0.05, (**) *p* < 0.01, (***) *p* < 0.001, (#) *p* < 0.05, (##) *p* < 0.01, (###) *p* < 0.001.

## 5. Conclusions

The optimal extraction method was obtained by the response surface method, and the most optimal conditions for ultrasonic-assisted extraction were an extraction temperature of 60 °C, an extraction time of 60 min, a solid–liquid ratio of 40 g/mL, and an ultrasonic power of 70 W. Five polysaccharides were isolated and purified from *S. vaninii* and named SVP-40, SVP-50, SVP-60, SVP-70, and SVP-80, respectively. Five kinds of polysaccharides were prepared by concentration, gradient alcoholysis, deproteinization, DEAE-52, anionic gel exchange chromatography, dialysis, and drying. The contents of the five polysaccharides were 85.53%, 86.90%, 92.40%, 90.12%, and 85.65%, respectively. The molecular weights of the five polysaccharides were 295.06 kDa, 215.36 kDa, 245.09 kDa, 273.75 kDa, and 274.77 kDa, respectively. All five polysaccharides had lower protein and nucleic acid contents, similar to the ultraviolet results. SVP-60 was composed of Man (1.4 mol%), Rham (1.1 mol%), GluA (0.5 mol%), GalA (0.6 mol%), Glu (2.0 mol%), and Ara (1.8 mol%). The FT-IR spectral analysis showed that all five polysaccharides had the basic polysaccharide structure. On this basis, the antioxidant activity of the five polysaccharides was studied in vitro. The in vitro antioxidant activity results show that the five polysaccharides showed good antioxidant activity in terms of DPPH, the superoxide anion scavenging ability, the hydroxyl radical scavenging ability, and the reducing ability. Preliminary studies have shown that the SVP-60 polysaccharide has different therapeutic effects on LPS-induced inflammatory responses in RAW.264.7 macrophages and Kupffer macrophages in vitro. SVP-60 significantly reduces levels of the pro-inflammatory factors IL-6, TNF-α, and IL-1β in RAW.264.7 cells, and H-SVP increases the level of the anti-inflammatory factor IL-10 in RAW.264.7 cells. This study provides a theoretical basis for further studies on the structure of the *S. vaninii* polysaccharide and the development of functional food or drugs, and it also provides the possibility for anti-inflammatory and antioxidant clinical applications. However, the structure and in vivo antioxidant and anti-inflammatory effects of polysaccharides deserve further investigation, and the functions and applications of polysaccharides also deserve further study to improve the utilization efficiency.

## Figures and Tables

**Figure 1 molecules-28-06081-f001:**
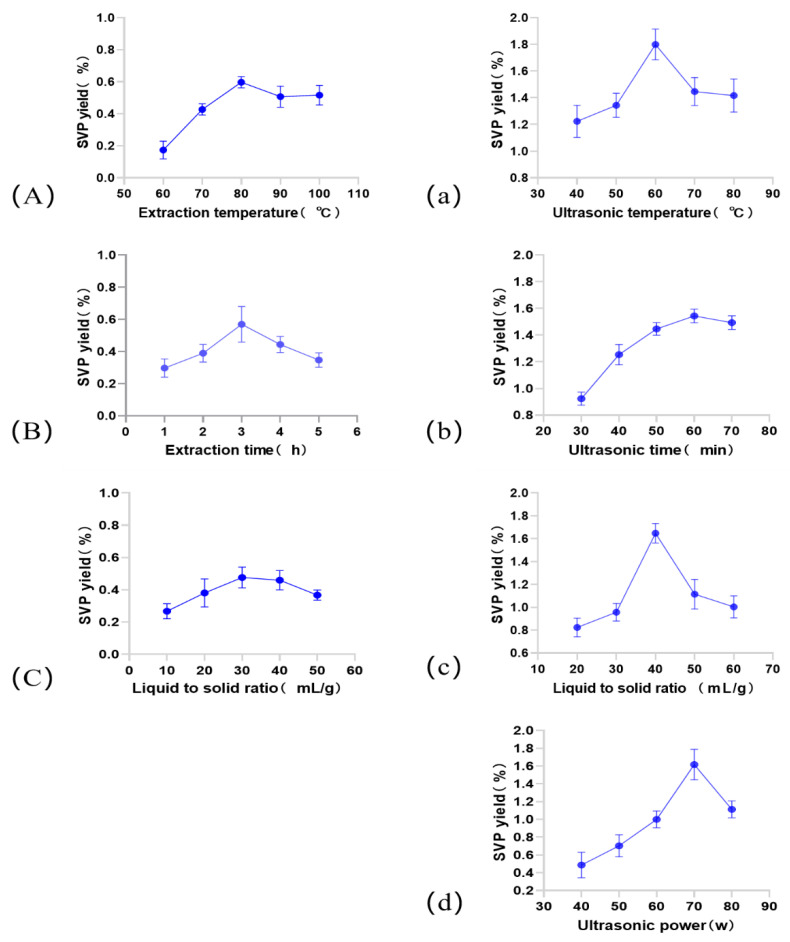
Effects of different extraction parameters on the SVP extraction yield. (**A**–**C**): Effects of the extraction temperature, extraction time and liquid–solid ratio on the extraction yield of SVP in the HWE process. (**a**–**d**): Effects of the ultrasonic temperature, ultrasonic time, liquid–solid ratio, and ultrasonic power on the extraction yield of SVP in the UAE process.

**Figure 2 molecules-28-06081-f002:**
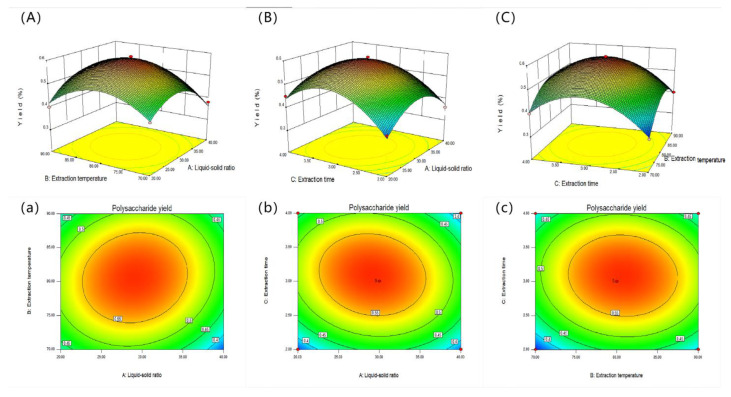
Response surface curves (2D and 3D) (**A**–**C**,**a**–**c**) showing the interactions of the liquid–solid ratio (**A**), extraction temperature (**B**), and extraction time (**C**) and their effects on the extraction yield of SVP in the HWE process.

**Figure 3 molecules-28-06081-f003:**
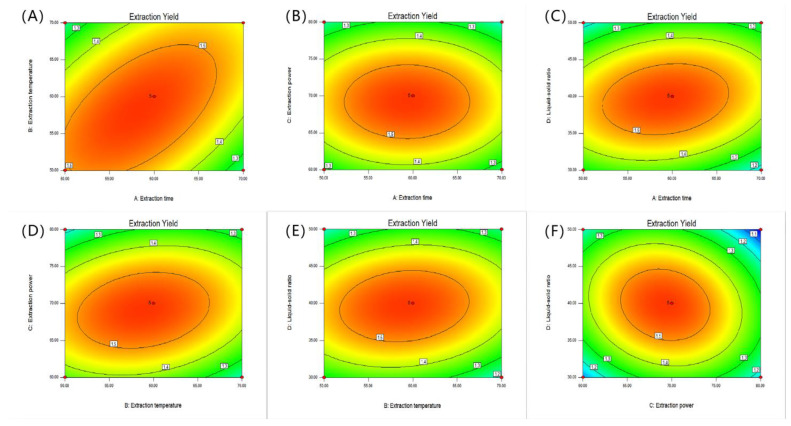
Response surface curves (2D) (**A**–**F**) showing the extraction rates of SVP in the UAE process for the extraction time (**A**), extraction temperature (**B**), ultrasonic power (**C**), and liquid–solid ratio (**D**).

**Figure 4 molecules-28-06081-f004:**
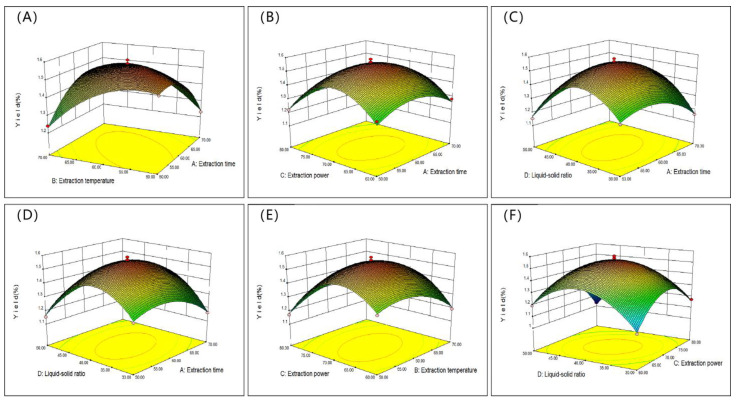
Response surface curves (3D) (**A**–**F**) showing the extraction rates of SVP in the UAE process for the extraction time (**A**), extraction temperature (**B**), ultrasonic power (**C**), and liquid–solid ratio (**D**).

**Figure 5 molecules-28-06081-f005:**
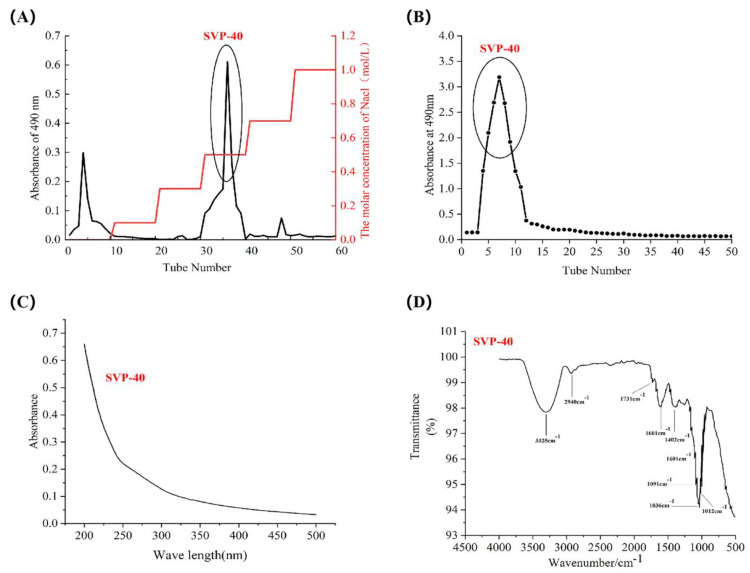
Elution curve for polysaccharide precipitated with 40% ethanol. (**A**) DEAE-52; (**B**) Superdex-100; (**C**) UV; (**D**) FT-IR analysis.

**Figure 6 molecules-28-06081-f006:**
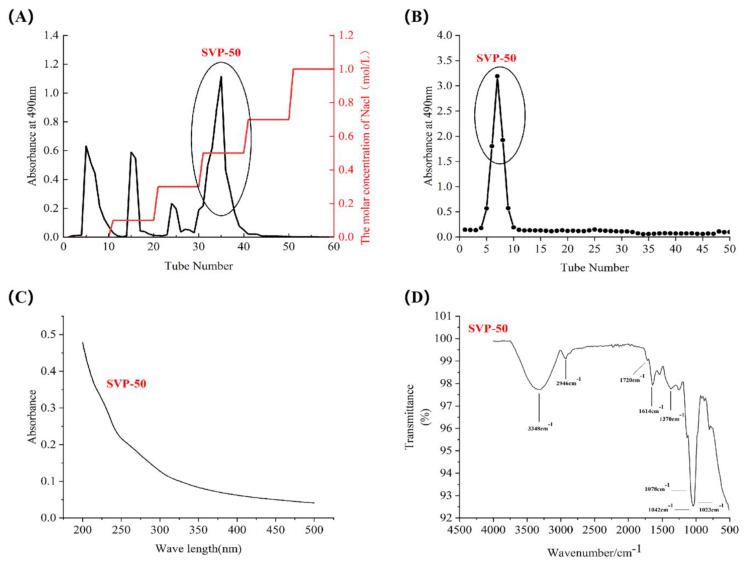
Elution curve for polysaccharide precipitated with 50% ethanol precipitated. (**A**) DEAE-52, (**B**) Superdex-100, (**C**) UV, (**D**) FT-IR analysis.

**Figure 7 molecules-28-06081-f007:**
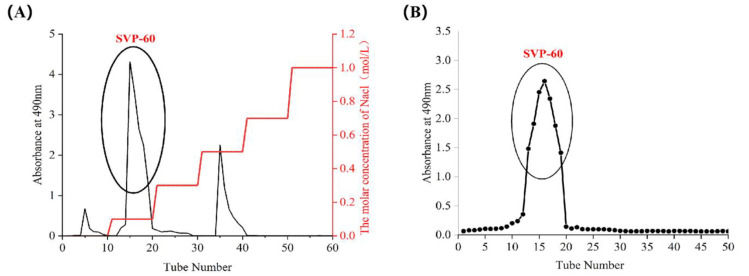
Elution curve for polysaccharide precipitated with 60% ethanol. (**A**) DEAE-52; (**B**) Superdex-100; (**C**) UV; (**D**) FT-IR analysis.

**Figure 8 molecules-28-06081-f008:**
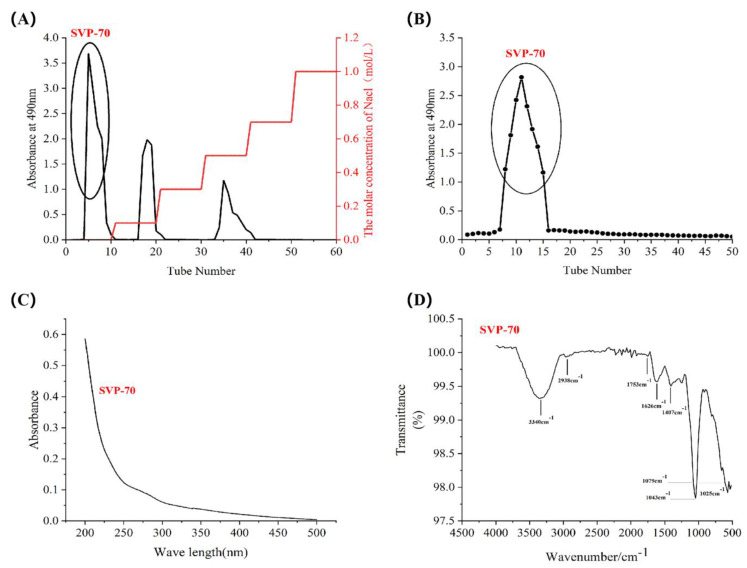
Elution curve for polysaccharide precipitated with 70% ethanol. (**A**) DEAE-52; (**B**) Superdex-100; (**C**) UV; (**D**) FT-IR analysis.

**Figure 9 molecules-28-06081-f009:**
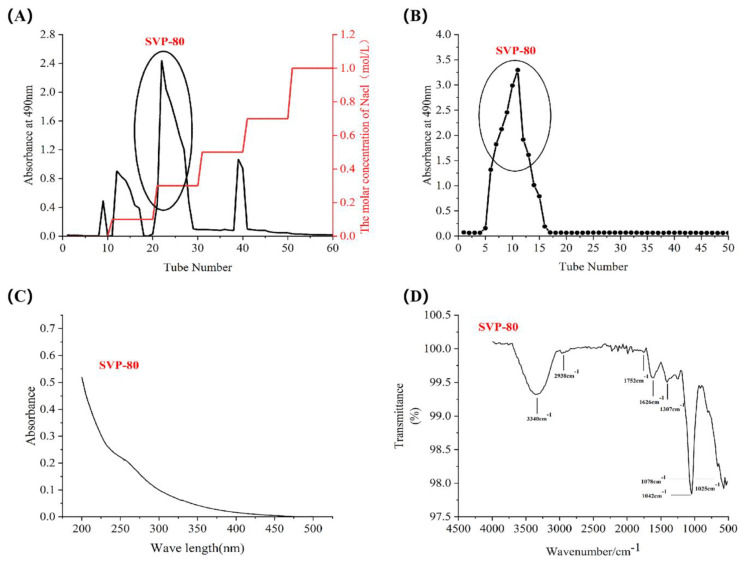
Elution curve for polysaccharide precipitated with 80% ethanol. (**A**) DEAE-52; (**B**) Superdex-100; (**C**) UV; (**D**) FT-IR analysis.

**Figure 10 molecules-28-06081-f010:**
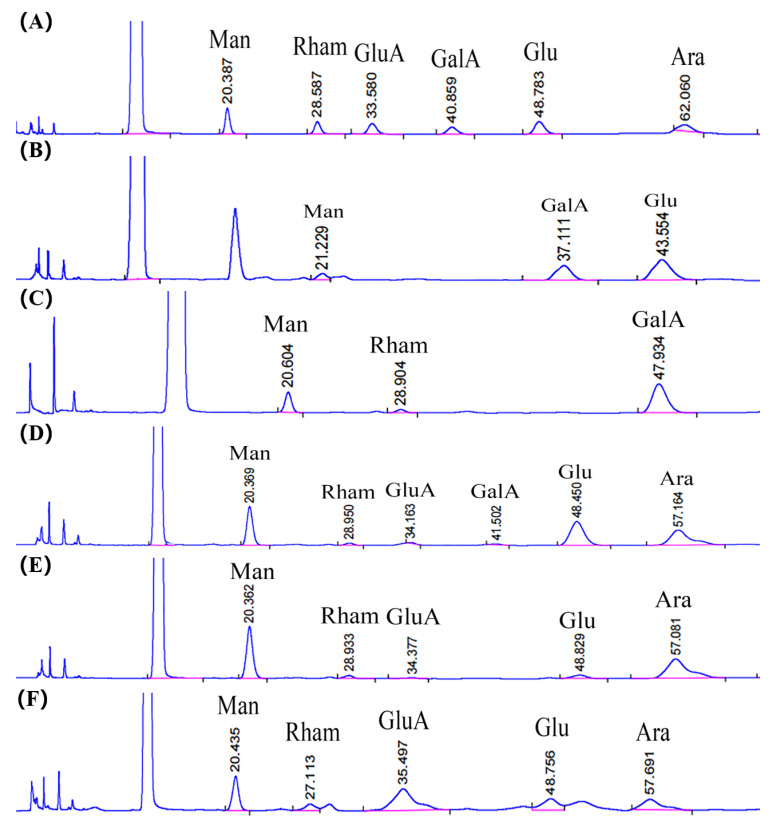
Schematic diagram of the monosaccharide compositions. (**A**) Standard curve of the monosaccharides; (**B**) monosaccharide composition of SVP-40; (**C**) monosaccharide composition of SVP-50; (**D**) monosaccharide composition of SVP-60; (**E**) monosaccharide composition of SVP-70; (**F**) monosaccharide composition of SVP-80. Acronyms: Man-D—Mannose, Rham-L—rhamnose monohydrate, GluA—glucuronic acid, GalA—Galacturonic acid, Glu—Glucose, Ara—arabinose.

**Figure 11 molecules-28-06081-f011:**
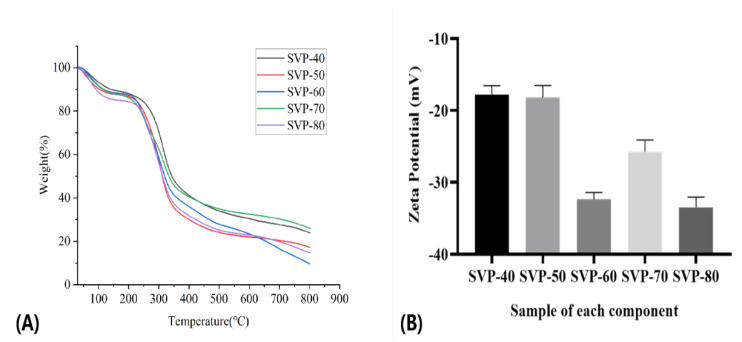
(**A**) Thermogravimetric analysis curves of SVP-40, SVP-50, SVP-60, SVP-70, and SVP-80. (**B**) Zeta potential analysis of SVP-40, SVP-50, SVP-60, SVP-70, and SVP-80.

**Figure 12 molecules-28-06081-f012:**
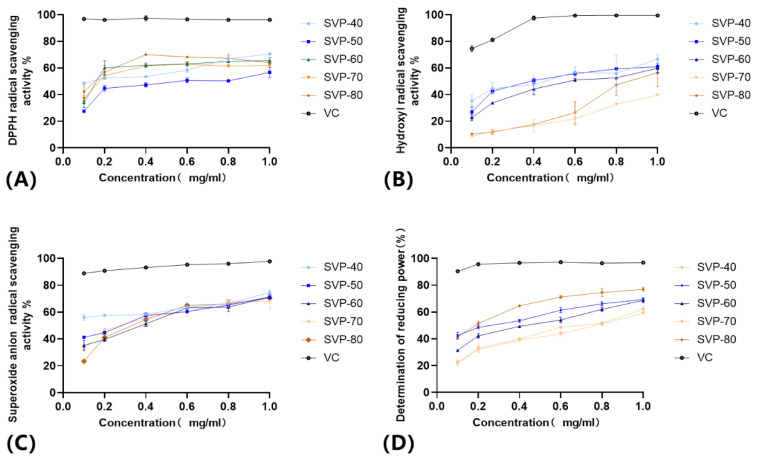
Determination of the antioxidant capacity of the five polysaccharides. (**A**) DPPH radical scavenging ability; (**B**) hydroxyl radical scavenging ability; (**C**) superoxide anion radical scavenging ability; (**D**) determination of the reducing power. Vitamin C was used as a positive control.

**Figure 13 molecules-28-06081-f013:**
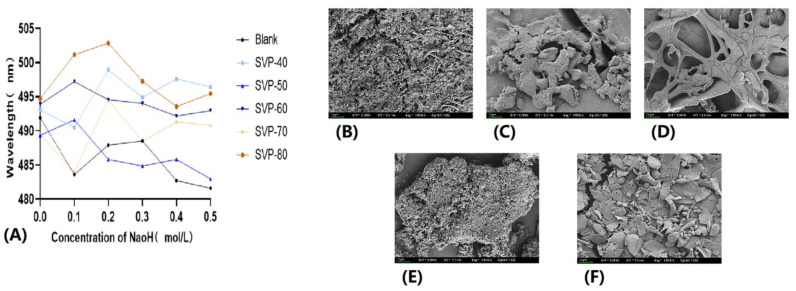
(**A**) Thermogravimetric analysis curves of SVP-40, SVP-50, SVP-60, SVP-70, and SVP-80. Scanning electron microscopy (SEM) images of five polysaccharides (**B**): SVP-40-100×, (**C**): SVP-50-100×, (**D**): SVP-60-100×, (**E**): SVP-70-100×, and (**F**): SVP-80-100×.

**Figure 14 molecules-28-06081-f014:**
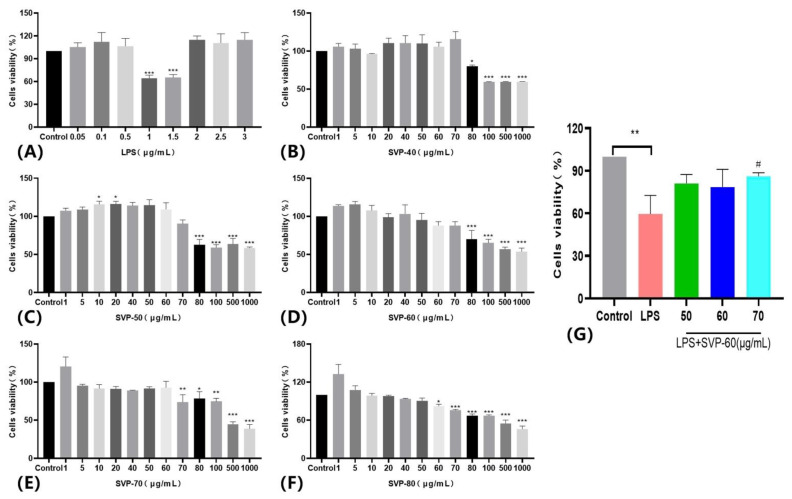
Cell viability of RAW 264.7. (**A**) The effect of LPS on the cell viability of RAW 264.7. (**B**) The effect of SVP-40 on the cell viability of RAW 264.7. (**C**) The effect of SVP-50 on the cell viability of RAW 264.7. (**D**) The effect of SVP-60 on the cell viability of RAW 264.7. (**E**) The effect of SVP-70 on the cell viability of RAW 264.7. (**F**) The effect of SVP-80 on the cell viability of RAW 264.7. (**G**) The effect of SVP-60 on cell viability of RAW 264.7 after LPS modeling. Data are represented as $\overset{-}{x}$ ± SD (*n* = 3). Compared with the control group, * *p* < 0.05, ** *p* < 0.01, *** *p* < 0.001. Compared with the model group, # *p* < 0.05.

**Figure 15 molecules-28-06081-f015:**
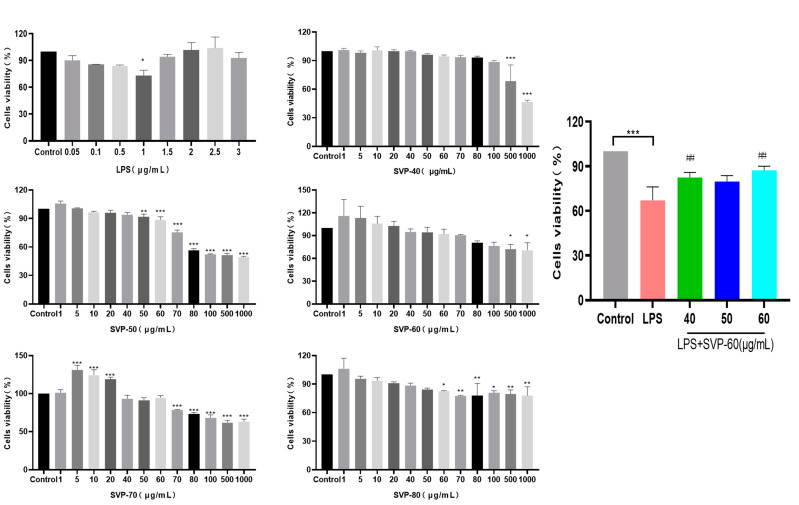
Cell viability of Kupffer cells. (**A**) The effect of LPS on the cell viability of Kupffer cells. (**B**) The effect of SVP-40 on the cell viability of Kupffer cells. (**C**) The effect of SVP-50 on the cell viability of Kupffer cells. (**D**) The effect of SVP-60 on the cell viability of Kupffer cells. (**E**) The effect of SVP-70 on the cell viability of Kupffer cells. (**F**) The effect of SVP-80 on the cell viability of Kupffer cells. (**G**) The effect of SVP-60 on cell viability of Kupffer cells after LPS modeling. Data are represented as $\overset{-}{x}$ ± SD (*n* = 3). Compared with the control group, * *p* < 0.05, ** *p* < 0.01, *** *p* < 0.001. Compared with the model group, ## *p* < 0.01.

**Figure 16 molecules-28-06081-f016:**
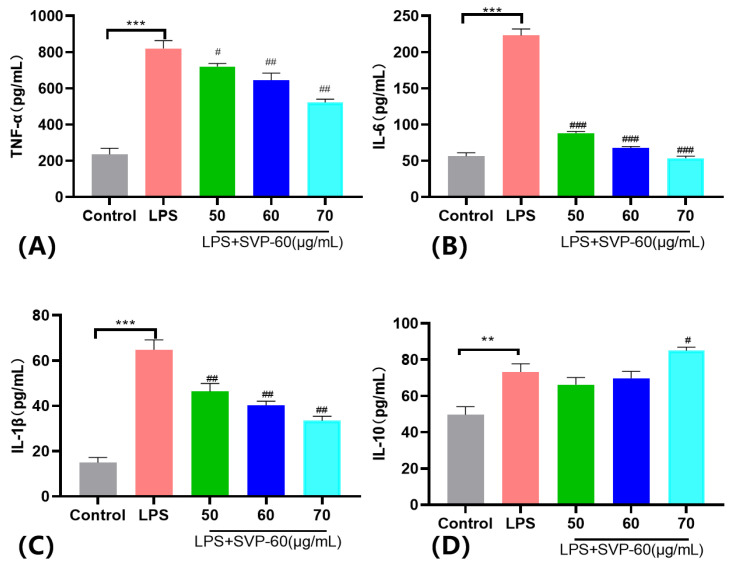
Effects of SVP-60 on (**A**) TNF-α, (**B**) IL-6, (**C**) IL-1β, and (**D**) IL-10 in LPS-induced RAW 264.7 cells. The cells were pretreated with different concentrations of SVP-60 for 1 h and then exposed to 1 μg/mL LPS for 24 h. The levels of TNF-α, IL-1β, IL-6, and IL-10 in the supernatant were determined by ELISA. Data show the mean  ±  SD of three independent experiments. Data are represented as $\overset{-}{x}$ ± SD (*n* = 3). Compared with the control group, ** *p* < 0.01, *** *p* < 0.001. Compared with the model group, # *p* < 0.05, ## *p* < 0.01, ### *p* < 0.001.

**Table 1 molecules-28-06081-t001:** The estimated regression estimates the relationship between the response variable (SVP production) and the independent variable of the HWE regression model.

Source	HWE					
	Sum of Squares	df	Mean Square	F Value	*p*-Value	Significant *
Model	0.14	9	0.016	61.69	<0.0001	**
A	0.002	1	0.002	5.87	0.0460	*
B	0.002	1	0.002	5.87	0.0460	*
C	0.003	1	0.003	12.41	0.0097	*
AB	0.001	1	0.001	4.75	0.0657	
AC	0.002	1	0.002	6.20	0.0415	*
BC	0.000	1	0.000	1.55	0.2530	
A^2^	0.036	1	0.036	140.47	<0.0001	**
B^2^	0.032	1	0.032	125.73	<0.0001	**
C^2^	0.051	1	0.051	198.48	<0.0001	**
R^2^ = 0.9875, R^2^adj = 0.9715, C.V.% = 3.56.						

Note: * <0.05 significant differences ** <0.01 very significant differences.

**Table 2 molecules-28-06081-t002:** The estimated regression estimates the relationship between the response variable (SVP production) and the independent variable of the UAE regression model.

Source	UAE					
	Sum of Squares	df	Mean Square	F Value	*p*-Value	Significant *
Model	0.64	14	0.046	125.97	<0.0001	**
A	0.002	1	0.002	6.13	0.0267	*
B	0.004	1	0.004	10.17	0.0066	**
C	0.012	1	0.012	33.30	<0.0001	**
D	0.002	1	0.002	4.98	0.0425	*
AB	0.053	1	0.053	146.38	<0.0001	**
AC	0.000	1	0.000	0.000	1.0000	
AD	0.012	1	0.012	32.57	<0.0001	**
BC	0.012	1	0.012	33.48	<0.0001	**
BD	0.007	1	0.007	15.56	0.0015	**
CD	0.014	1	0.014	39.85	<0.0001	**
A^2^	0.069	1	0.069	192.27	<0.0001	**
B^2^	0.062	1	0.062	171.06	<0.0001	**
C^2^	0.28	1	0.28	764.43	<0.0001	**
D^2^	0.33	1	0.33	906.61	<0.0001	**
R^2^ = 0.9921, R^2^adj = 0.9842, C.V.% = 1.48.						

Note: * <0.05 significant differences ** <0.01 very significant differences.

**Table 3 molecules-28-06081-t003:** Yield, chemical composition, monosaccharide composition, monosaccharide molar ratio, and molecular weight of each gradient ethanol precipitated polysaccharide.

Samples	SVP-40	SVP-50	SVP-60	SVP-70	SVP-80
Yield (%)	0.35%	0.39%	0.77%	0.26%	1.08%
Total sugar content (%)	85.53%	86.90%	92.40%	90.12%	85.65%
Protein (%)	0.29%	0.19%	0.91%	2.43%	0.32%
Uronic acid (%)	6.03%	7.87%	4.39%	0.95%	0.47%
Monosaccharide composition (mol %)					
Man (Mannose)	2.2	5.5	1.4	1.9	1.1
Rham (Rhamnose)	_	1.2	1.1	1.5	3.5
GluA (Glucuronic acid)	_	_	0.5	0.8	2.3
GalA (Galactose acid)	8.5	1.8	0.6	_	_
Glu (Glucose)	13.7	_	2.0	2.7	1.0
Ara (Arabinose)	_	_	1.8	2.3	1.0
Molecular weight (k Da)	295.06	215.36	245.09	273.75	274.77

**Table 4 molecules-28-06081-t004:** Box–Behnken experimental design and results for the yield of the SVP in the HWE process.

Test Group	Coded Levels			Response Value	
	A Liquid-to-Solid Ratio (mL/g)	B Extraction Temperature (℃)	C Extraction Time (h)	Measured Value (%)	Predictive Value (%)
1	40.00 (1)	70.00 (−1)	3.00 (0)	0.38	0.36
2	30.00 (0)	80.00 (0)	3.00 (0)	0.58	0.59
3	30.00 (0)	90.00 (1)	4.00 (1)	0.42	0.41
4	20.00 (−1)	80.00 (0)	4.00 (1)	0.45	0.44
5	40.00 (1)	80.00 (0)	2.00 (−1)	0.36	0.37
6	20.00 (−1)	80.00 (0)	2.00 (−1)	0.37	0.36
7	30.00 (0)	80.00 (0)	3.00 (0)	0.60	0.59
8	30.00 (0)	90.00 (1)	2.00 (−1)	0.40	0.39
9	30.00 (0)	70.00 (−1)	2.00 (−1)	0.34	0.35
10	30.00 (0)	80.00 (0)	3.00 (0)	0.58	0.59
11	30.00 (0)	70.00 (−1)	4.00 (1)	0.40	0.41
12	40.00 (1)	80.00 (0)	4.00 (1)	0.36	0.37
13	20.00 (−1)	90.00 (1)	3.00 (0)	0.40	0.42
14	30.00 (0)	80.00 (0)	3.00 (0)	0.58	0.59
15	30.00 (0)	80.00 (0)	3.00 (0)	0.60	0.59
16	20.00 (−1)	70.00 (−1)	3.00 (0)	0.42	0.42
17	40.00 (1)	90.00 (1)	3.00 (0)	0.43	0.42

**Table 5 molecules-28-06081-t005:** Box–Behnken experimental design and results for the yield of the SVP in the UAE process.

Test Group	Coded Levels				Response Value	
	A Extraction Time (min)	B Extraction Temperature (℃)	C Liquid-to-Solid Ratio (mL/g)	D Ultrasound Power (W)	Measured Value (%)	Predictive Value (%)
1	70.00 (1)	60.00 (0)	70.00 (0)	50.00 (1)	1.24	1.25
2	60.00 (0)	60.00 (0)	70.00 (0)	40.00 (0)	1.58	1.55
3	60.00 (0)	60.00 (0)	70.00 (0)	40.00 (0)	1.54	1.55
4	60.00 (0)	70.00 (1)	80.00 (1)	40.00 (0)	1.24	1.25
5	60.00 (0)	70.00 (1)	70.00 (0)	50.00 (1)	1.24	1.24
6	60.00 (0)	60.00 (0)	70.00 (0)	40.00 (0)	1.54	1.55
7	70.00 (1)	60.00 (0)	80.00 (1)	40.00 (0)	1.19	1.19
8	50.00 (−1)	60.00 (0)	70.00 (0)	30.00 (−1)	1.29	1.30
9	70.00 (1)	60.00 (0)	60.00 (−1)	40.00 (0)	1.28	1.26
10	60.00 (0)	60.00 (0)	80.00 (1)	50.00 (1)	1.03	1.01
11	60.00 (0)	50.00 (−1)	60.00 (−1)	40.00 (0)	1.34	1.35
12	50.00 (−1)	60.00 (0)	80.00 (1)	40.00 (0)	1.22	1.22
13	70.00 (1)	70.00 (1)	70.00 (0)	40.00 (0)	1.44	1.43
14	70.00 (1)	50.00 (−1)	70.00 (0)	40.00 (0)	1.23	1.24
15	50.00 (−1)	60.00 (0)	60.00 (−1)	40.00 (0)	1.31	1.29
16	60.00 (0)	50.00 (−1)	70.00 (0)	30.00 (−1)	1.31	1.29
17	60.00 (0)	60.00 (0)	70.00 (0)	40.00 (0)	1.56	1.55
18	50.00 (−1)	50.00 (−1)	70.00 (0)	40.00 (0)	1.49	1.49
19	60.00 (0)	50.00 (−1)	80.00 (1)	40.00 (0)	1.17	1.18
20	60.00 (0)	60.00 (0)	70.00 (0)	40.00 (0)	1.53	1.55
21	60.00 (0)	70.00 (1)	60.00 (−1)	40.00 (0)	1.19	1.21
22	70.00 (1)	60.00 (0)	70.00 (0)	30.00 (−1)	1.16	1.17
23	60.00 (0)	60.00 (0)	60.00 (−1)	30.00 (−1)	1.09	1.10
24	50.00 (−1)	70.00 (1)	70.00 (0)	40.00 (0)	1.24	1.23
25	60.00 (0)	60.00 (0)	60.00 (−1)	50.00 (1)	1.19	1.20
26	60.00 (0)	70.00 (1)	70.00 (0)	30.00 (−1)	1.19	1.18
27	50.00 (−1)	60.00 (0)	70.00 (0)	50.00 (1)	1.15	1.17
28	60.00 (0)	50.00 (−1)	70.00 (0)	50.00 (1)	1.21	1.20
29	60.00 (0)	60.00 (0)	80.00 (1)	30.00 (−1)	1.17	1.16

## Data Availability

The research data used to support the findings of this study are included in the article.

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
