# Peer review of "Extraction, Purification, and Structural Characterization of Polysaccharides from Sanghuangporus vaninii with Anti-Inflammatory Activity"

_molecules, 2023, doi:10.3390/molecules28166081_

Round 1

Reviewer 1 Report

Please, see attached document.

Author Response

Response to Reviewer 1 Comments

We appreciate it very much for these good suggestions, and we have done it according to your ideas.We have also made changes to the introduction, and provided sufficient background and included all relevant references.

Point 1: Line 16: Give full meaning of “SVA” and “TG” in line 19. Give full meanings of abbreviations the first time mentioned. Please, do so for other abbreviations.

Response 1: We are very sorry for our unclear standard abbreviations and thank you for your valuable comments. It has been modified in the original text and marked in red.

Sanghuangporus vaninii -(S. vaninii)

SVP- (Sanghuangporus vaninii polysaccharides)

TG- (Thermogravimetric analysis)

Zeta-(Zeta potential )

SEM -(Scanning electron microscope)

RSM-(Response surface method )

FT-IR - (Fourier Transform infrared spectral )

HPLC - (High-Performance Liquid Chromatography)

Man - (Mannose)

GluA -(Glucuronic acid)

Glu -(Glucose)

GalA -(Galactose acid)

Ara-(Arabinose)

Rham - (Rhamnose)

Point 2:Line 20: Delete “to” from the sentence and replace “investigate” with “investigated

Response 2: For our mistake, we feel very sorry, for your precious opinions thank you very much. At present, we have modified the original text and marked it in red.

Point 3:More recent literature/references on this topic is welcome.

Response 3: We are very grateful for these suggestion raised by the reviewer. Our paper have been modified and marked in red in introduction.

Point 4:Line 32: “daai” Please, check correct spelling

Rsponse 4: We are very sorry for our mistake. In addition, "daai" has been modified to Y.C.Dai in the original and marked in red.

Point 5:Lines 37- 39: The statement is not clear, if these are referring to findings relating to biological activities of Sanghuangporus vaninii, Please, kindly rephrase the sentence to bring out major findings and references. Similarly, if used for drugs and health products, what type and functionality

Response 5: Thank you for your nice suggestion.We have corrected the statement section, revised the main findings of the biological activities of Sanghuangporus vaninii and the references section, and discussed the related drugs and nutriceups of Sanghuangporus vaninii.And it has been marked in red in the original.

Point 6:Line 90: “……a phenomenon identical to that report by some” Please, show what was done and insert references.

Response 6: We feel sorry for the inconvenience brought to the reviewer. References have been added to the original and marked in red.

Point 7:Line 461: Mention/indicate the particular fungus harvested.

Response 7: Thank you so much for your careful check.We have indicated the particular fungus harvested.And it has been marked in red in the original.

Point 8:Line 468: Please, specify quantity of material/solvent used.

Response 8: Thank you so much for your careful check.The amount of material/solvent used is specified. And it has been marked in red in the original.

We feel sorry for our carelessness.

Point 9:Line 591: “Each dose is placed in three parallel holes”? Is the experiment carried out in triplicates? If so, please, report appropriately

Response 9: We totally understand the reviewer‘s concern. Each dose is placed in three parallel holes.This article refers to three repetitions per dose.We feel sorry for our carelessness.

Reviewer 2 Report

This is a very long biophysical characterized and bioassays based analyzed manuscript on various fraction of polysaccharides. Its enormous amount of work.

Abstract:

This sentence in the abstract need to be modified “ Our study would provide theoretical support for potential applications of Sanghuangporus vaninii 26 during traditional use and the health care fields.”

Introduction: Italicie all the botanical names “(S. vaninii)”.

Longer sentences need to be divided into smaller ones for clarity.

The authors should compare the bioassays with the available literature on polysaccharides from similar plants.

Should be improved

Author Response

Response to Reviewer 2 Comments

We appreciate it very much for this good suggestion. We have made changes to background,relevant references design and results of the research. We regret there were problems with the English. The paper has been carefully revised by a professional language editing service to improve the grammar and readability.

Point 1: .Abstract:This sentence in the abstract need to be modified “ Our study would provide theoretical support for potential applications of Sanghuangporus vaninii 26 during traditional use and the health care fields.

Response 1: We gratefully appreciate for your valuable suggestion. We have changed in line 57-58, page 2, and marked it in red.

Point 2:Introduction: Italicie all the botanical names “(S. vaninii)”.

Response 2: Thank you so much for your careful check..We have changed this error.We feel sorry for our carelessness.

Point 3:Longer sentences need to be divided into smaller ones for clarity.

Response 3: We totally understand the reviewer‘s concern. We have corrected the error and marked it in red throughout the text.

Point 4:The authors should compare the bioassays with the available literature on polysaccharides from similar plants.

Response 4:We have added in the bioassay a comparison of the existing literature regarding similar plant polysaccharides and labelled them using the colour red.

Point 5:Comments on the Quality of English Language:Should be improved

Response 5:We apologise for any problems with the English language.This article has been carefully revised by a professional language editing service to improve grammar and readability.

Reviewer 3 Report

The authors extracted polysaccharide (SVP) from Sanghuangporus vaninii and characterized them using high-performance liquid chromatography, ultraviolet spectrum, Fourier transform infrared spectrum, TG, Zeta and SEM. Also, the types of constituent sugars were analyzed.

In addition, antioxidant activity was examined, and anti-inflammatory activity was analyzed using two types of cell lines.

However, for the following reasons, I judge that it is difficult to publish in Molecules in the current state.

1. The introductory part is too weak.

Please add the recent papers reported so far by plant-derived polysaccharides. Also, explain why the authors paid attention to the anti-inflammatory activity despite the fact that many papers have reported on the enhancement of innate immunity by plant polysaccharides.

2. Research results on anti-inflammatory activity are poor.

The authors showed that several cytokines are inhibited by SVP in LPS-stimulated RAW264.7 cells. However, data confirming the phosphorylation of several intracellular signaling pathway proteins (MAPKs, NF-kB, Akt) are also considered essential.

Author Response

Response to Reviewer 3 Comments

We appreciate it very much for this good suggestion. We have made changes to background,relevant references, design,result,methods and conclusion of the research.

Point 1: The introductory part is too weak. Please add the recent papers reported so far by plant-derived polysaccharides. Also, explain why the authors paid attention to the anti-inflammatory activity despite the fact that many papers have reported on the enhancement of innate immunity by plant polysaccharides.

Response 1: We gratefully appreciate for your valuable suggestion. And we have made the appropriate changes in the original text in accordance with your comments, marked in red. Firstly, we have modified the introductory part in the original text. Secondly, we have added recent reports on polysaccharides in the introduction and discussion parts of the paper.Thirdly, we clearly understand that current reports focus on polysaccharides to enhance innate immunity. However, this thesis remains focused on the anti-inflammatory activity of polysaccharides for the following reasons. We investigated the anti-inflammatory activity of SVP-60 on two types of macrophages RAW. 264.7 and Kupffer cells. Both macrophages are besides used as one of the models of immune response. One of them, RAW. 264.7 macrophages, can also be used as a model for polysaccharide treatment of intestinal inflammation, and Kupffer cells can also be used as a model for polysaccharide treatment of liver inflammation. Therefore, this experiment can study the inflammatory response. The ultimate goal of this project is to demonstrate that SVP-60 has the treatment of liver inflammation and intestinal inflammation. The current study of the anti-inflammatory effect is the basis for the follow-up of the thesis research.

Point 2: Research results on anti-inflammatory activity are poor.The authors showed that several cytokines are inhibited by SVP in LPS-stimulated RAW264.7 cells.   However, data confirming the phosphorylation of several intracellular signaling pathway proteins (MAPKs, NF-kB, Akt) are also considered essential.

Response 2: Thank you for this valuable feedback. We agree that more studies or more data would be useful for the study. We understand that phosphorylation of several intracellular signalling pathway proteins (MAPKs, NF-kB, Akt) may better reveal anti-inflammatory pathways. However, in this study, we focused on how Sanghuangporus vaninii increased polysaccharide yield while also structurally characterising its five different fractions, also confirming their potential application in antioxidant and anti-inflammatory fields. We believe that the inhibition of several anti-inflammatory and pro-inflammatory cytokines by Sanghuangporus vaninii in LPS-stimulated RAW264.7 cells may not be optimal, but it should be sufficient to conclude that Sanghuangporus vaninii has an inhibitory effect on the inflammation induced by LPS in RAW264.7 cells.

And we know from extensive literature reading,in "Composition and anti-inflammatory effect of polysaccharides from Sargassum horneri in RAW264.7 macrophages", only the levels of NO, TNF-α and IL-7 production secreted by LPS-stimulated RAW264.7 macrophages were determined to have strong anti-inflammatory effects, and there was no mention of the related intracellular signalling pathway proteins (https://doi.org/10.1016/j.ijbiomac.2016.02.025). Anti-inflammatory constituents from Perilla frutescens on lipopolysaccharide-stimulated RAW264.7 cells were identified in the publication’ by Wang et al,the levels of NO, TNF-α and IL-6 secretion in LPS-stimulated RAW264.7 cells were identified as indicators of their anti-inflammatory activity. (https://doi.org/10.1016/j.fitote.2018.08.006). Wang et al published "Phytochemicals from Anneslea fragrans Wall and their Hepatoprotective and Anti-Inflammatory Activities" ,RAW264.7 cells released high levels of pro-inflammatory cytokines, including IL-6, IL-1β, and TNF-α, then the author believes that certain in vitro anti-inflammatory activity (https://doi.org/10.3390/molecules28145480).

In additon ,We are investigating the in vivo anti-inflammatory activity of SVP-60 in mouse liver and intestine. Unfortunately, the results are not yet available, and we are adding the anti-inflammatory part of the discussion, which is sufficiently labelled in red.    

Reviewer 4 Report

1.      The results were not inconsistent with the statement in line 316 – 317, page 13. SVP-40 only had the higher contents of Glu and GalA. Please provide the DPPH radical scavenging capacity of Glu and GalA.

2.      Please delete the statement “is closest to Vc”. The effect of any group has a large gap with Vc.

3.      There are no differences on the DPPH radicals scavenging ability and Superoxide anion radicals scavenging ability among all SVPs. Please correct the related description.

4.      Please explain the different effects of SVP-60 in LPS-treated RAW264.7 Cells and Kupffer cells after LPS treatment.

5.      Please explain the different effects of SVP-60 on the viability, pro-Inflammatory and anti-in-439 flammatory factor in LPS-treated RAW264.7 Cells.

6.      Please recheck figure 14.

7.      Please check the statement in line 57 – 58, page 2. This statement does not fully correspond to the results in Figure 1A.

8.      Please check the statement in line 90 – 92, page 2.

Author Response

.

Response to Reviewer 4 Comments

We appreciate it very much for this good suggestion, and we have done it according to

your ideas. We have also made changes to the research design,result,methods and conclusion.

Point 1: The results were not inconsistent with the statement in line 316 – 317, page 13.SVP-40 only had the higher contents of Glu and GalA. Please provide the DPPH radical scavenging capacity of Glu and GalA.

Response 1: Thank you for your valuable suggestion, unfortunately we are unable to provide the DPPH free radical scavenging capacity of Glu and GalA. The present experiments provide the antioxidant activity of SVP-40, and despite the high content of Glu and GalA in it, it still does not indicate that the other monosaccharides that make up SVP-40 have no effect. Therefore, we believe that it is more convincing to provide the DPPH activity of total sugars in SVP-40.In the manuscript, after checking, we believe that there is some ambiguity in the reference to the higher DPPH free radical scavenging capacity of SVP-40 being related to the levels of Glu and GalA in 2.10.1. This has now been removed.

Point 2:Please delete the statement “is closest to VC”. The effect of any group has a large gap with VC.

Response 2: .Thank you so much for your careful check. we agree with the point that the effect of any group has a large gap with VC. We are sorry that we did not do a statistical comparison at the same concentration initially. At your suggestion, we did a statistical comparison and there is indeed a gap. The error has now been removed.

Point 3:There are no differences on the DPPH radicals scavenging ability and Superoxide anion radicals scavenging ability among all SVPs. Please correct the related description.

Response 3: We gratefully appreciate for your valuable suggestion.At your suggestion, we have checked the descriptions related to SVP DPPH free radical and superoxide anion radical scavenging capacity and indeed there is no differences. We have now corrected the descriptions related to SVP DPPH radical and superoxide anion radical scavenging capacity in the manuscript.

Point 4:Please explain the different effects of SVP-60 in LPS-treated RAW264.7 Cells and Kupffer cells after LPS treatment.

Response 4:We explained the different effects of SVP-60 on RAW264.7 cells and Kupffer cells after LPS treatment and highlighted the results in red in 2.13.2.

Point5:Please explain the different effects of SVP-60 on the viability, pro-Inflammatory and anti-in-439 flammatory factor in LPS-treated RAW264.7 Cells.

Response 5:Thank you so much for your suggestions. In 2.13.3, we explained the different effects of SVP-60 on LPS- treated RAW264.7 cell activity, pro-inflammatory factors and anti-inflammatory factors in the red section.

Point 6: Please recheck figure 14.

Response 6:Thank you very much for your suggestion. We rechecked figure 14.We found an error in the image resolution and the content of the image and have changed this error

Point 7: Please check the statement in line 57 – 58, page 2. This statement does not fully correspond to the results in Figure 1A.

Response 7:At your suggestion, we have examined the note on page 2, lines 57 - 58, and believe that the presentation is indeed not entirely consistent with the results in Figure 1A. In the original text we have made changes and marked them in red.

Point 8:Please check the statement in line 90 – 92, page 2.

Response 8:We have changed the description of liquid-solid ratio of polysaccharide in 2.1.3 in red , thank you for your suggestion.

Round 2

Reviewer 3 Report

The authors addressed all issues.